# Efficacy of Non-Invasive Brain Stimulation for Refractory Obsessive-Compulsive Disorder: A Meta-Analysis of Randomized Controlled Trials

**DOI:** 10.3390/brainsci12070943

**Published:** 2022-07-19

**Authors:** Shu Zhou, Yan Fang

**Affiliations:** 1The Fourth School of Clinical Medicine, Zhejiang Chinese Medical University, Hangzhou 310053, China; zhoushu@zcmu.edu.cn; 2Department of Physiology, Zhejiang Chinese Medical University, Hangzhou 310053, China

**Keywords:** refractory obsessive-compulsive disorder, non-invasive brain stimulation, repetitive transcranial magnetic stimulation, theta-burst stimulation, transcranial direct current stimulation, meta-analysis

## Abstract

Obsessive-compulsive disorder (OCD) is a neuropsychiatric disorder, with 30–40% of OCD patients being unresponsive to adequate trials of anti-OCD drugs and cognitive behavior therapy. The aim of this paper is to investigate the efficacy of non-invasive brain stimulation (NIBS) on treating refractory OCD. With PubMed, Embase, PsycInfo, and Cochrane Library used on 15 February 2022, 24 randomized controlled trials involving 663 patients were included. According to this analysis, NIBS including repetitive transcranial magnetic stimulation (rTMS), theta-burst stimulation (TBS), and transcranial direct current stimulation (tDCS), had a moderate effect on the reduction of Yale-Brown Obsessive Compulsive Scale (Y-BOCS) scores (SMD = 0.54, 95% CI: 0.26–0.81; *p* < 0.01). In the subgroup analysis, rTMS seemed to produce a better therapeutic effect (SMD = 0.73, 95% CI: 0.38–1.08; *p* < 0.01). Moreover, excitatory (SMD = 1.13, 95% CI: 0.24–2.01; *p* = 0.01) and inhibitory (SMD = 0.81, 95% CI: 0.26–1.36; *p* < 0.01) stimulation of the dorsolateral prefrontal cortex (DLPFC) both alleviated OCD symptoms. In the secondary outcome of clinical response rates, NIBS treatment led to an increase in response rates (RR = 2.26, 95% CI: 1.57–3.25; *p* < 0.01).

## 1. Introduction

Obsessive-compulsive disorder (OCD) is a mental illness characterized by obsessions (frequent and persistent thoughts) and compulsions (repetitive actions or mental activities) [1]. The lifetime prevalence estimates for OCD are 2–3% [2,3]. OCD was associated with impaired quality of life [4,5] and increased mortality risk [6]. The first-line treatment options for OCD include selective serotonin reuptake inhibitor (SSRI) and cognitive behavior therapy (CBT) [7,8,9,10]. Unfortunately, 30–40% of individuals could not react to first-line therapies effectively [11]. At present, non-invasive brain stimulation (NIBS) can be performed to create or change neural plasticity in the nervous system [12].

Repetitive transcranial magnetic stimulation (rTMS) provides a non-invasive solution to inducing excitability changes in the cerebral cortex through the use of a wire that generates a magnetic field [13]. Low-frequency (LF, <1 Hz) stimulation is believed to suppress cortical excitability, and high-frequency (HF, >5 Hz) stimulation promotes cortical excitability [14]. Theta-burst stimulation (TBS) represents a new type of TMS that relies on the continuous (cTBS) or intermittent (iTBS) stimulation of the cortex to elicit inhibition or excitation [15]. TBS has the advantage of being more acceptable than other NIBS due to its short duration [16]. Transcranial direct current stimulation (tDCS) is a low-cost, simple-to-use NIBS with a high level of tolerability. It provides a mild direct current (1–2 mA) for a set number of minutes through two big electrodes attached to the scalp. The mechanism of action of tDCS on neuronal membranes has been proposed as follows: cathodal stimulation hyperpolarized neurons, while anodal stimulation depolarized them [17].

Evidence suggests that cortico-striato-thalamo-cortical (CSTC) circuits play a role in the neurological underpinnings of OCD [18]. Recent neurophysiological and neuroimaging investigations have revealed that OCD affects the function of the dorsolateral prefrontal cortex (DLPFC), orbitofrontal cortex (OFC), supplementary motor area (SMA), etc. [19,20,21]. Currently, there are several types of NIBS made available for the treatment of refractory OCD, but the effective type and optimal stimulation protocols are still debatable. In this meta-analysis, a comparison between three types of NIBS can provide better guidance on clinical selection.

## 2. Methods

### 2.1. Search Strategy and Eligibility Criteria

The Cochrane Handbook was followed to conduct this meta-analysis [22]. We used PubMed, Embase, PsycInfo, and Cochrane Library on 15 February 2022 to search the literature. This meta-analysis was registered with the PROSPERO under the registration number CRD42022315026.

The following terms were used in the search: (“obsessive-compulsive disorder” or “OCD”) and (“rTMS” or “TBS” or “tDCS”). To ensure more definitive conclusions, this research adopted stringent inclusion criteria as follows: Patients with a diagnosis of refractory OCD. Refractory OCD is defined as the failed adequate trials of anti-OCD drugs such as SSRIs.NIBS (including rTMS, TBS, tDCS) as a way of treatment for OCD.No change in the original medication regimen during the experiment.Trials that are randomized controlled trials, with a parallel or crossover design are used.Experiments that involve at least 10 stimulation sessions.The Yale-Brown Obsessive Compulsive Scale (Y-BOCS) change score or clinical response rate is available.

### 2.2. Data Extraction

We collected the following characteristics from the sample: the relevant literature (author, year of publication), the sample size, the age, the number of sessions, and the duration of a trial. The following parameters of therapy were retrieved for rTMS: the coil location, the frequency (Hz) of stimulation, the intensity (% of the resting motor threshold (rMT)) of stimulation, and the number of pulses delivered in each session. For TBS, the type of TBS was also collected (iTBS or cTBS). For tDCS, the location of the anode and cathode, the current strength (mA), and the session duration (min) were collected. The following data were retrieved for analysis: the Y-BOCS baseline and endpoint (the first measurement after finishing the treatment) scores, the Y-BOCS change score, and the clinical response rate. The data in the graph was obtained by using GetData software (version 2.5). For continuous data, the mean and standard deviation (SD) must be extracted.

### 2.3. Outcome Measures

The primary outcome was the Y-BOCS change score obtained after the completion of treatment. If no change score was reported in the original text, it will be calculated by using baseline and endpoint Y-BOCS scores. The secondary outcome was the clinical response rate. However, the definition of the clinical response varies, with most of the literature defining it as a >25% or >35% reduction in Y-BOCS score.

### 2.4. Data Analysis

The statistical analyses were conducted using the Stata software (version 16.0) in line with the Cochrane Handbook [22]. To assess the effect size of the research, Hedge’s g was calculated for the change in the Y-BOCS score. As for the standardized mean difference (SMD), its effect value of <0.40 indicates a small effect, its effect value of 0.40 to 0.70 indicates a moderate effect, and its effective value of >0.70 indicates a large effect. For the dichotomous outcome data on the number of respondents qualified for the response criteria, risk ratio (RR) was utilized to determine the total effect. RR > 1 is considered to be a preference for active treatment. On the contrary, RR < 1 indicates a preference for sham treatment. Heterogeneity was assessed using Q-test and metric I^2^. When *p* < 0.1 on the Q-test, it indicates the existence of heterogeneity. It is suggested that I^2^ ranges from 0% to 40%, 30% to 60%, 50% to 90%, and 75% to 100% may mean little, moderate, substantial, and considerable heterogeneity respectively. Egger’s regression analysis was conducted to examine publication bias. The analyses of subgroups were conducted on the following categories: (1) types of NIBS; and (2) different stimulation protocols. All of the above analyses adopted random effect models.

### 2.5. Risk of Bias Assessment

Literature quality was evaluated by two researchers, with the risk of bias assessed using the Cochrane tool. The quality of the selected literature was evaluated through seven items, including random sequence generation, allocation concealment, blinding of participants and personnel, blinding of outcome assessment, incomplete outcome data, selective reporting, and other biases. The above-mentioned items can be divided into three categories: low risk, unclear, and high risk. In the course of the literature evaluation, evaluators could discuss with each other to make final decisions in case of uncertain problems.

## 3. Results

### 3.1. Study Selection and Characteristics

This literature search is described in Figure 1, with 24 eligible studies identified. There were 18 rTMS studies [23,24,25,26,27,28,29,30,31,32,33,34,35,36,37,38,39,40], 3 TBS studies [41,42,43], and 3 tDCS studies [44,45,46]. A total of *n* = 663 individuals were included in the study, of which *n* = 346 received NIBS and *n* = 317 received sham stimulation. If studies adopted a crossover design, only the first phase data could be included to avoid legacy effects. Since some studies cannot obtain complete Y-BOCS change scores (including mean and SD) or clinical response rates, two studies [24,37] were excluded from the primary outcome analysis, and four studies [23,30,34,36] were excluded from the secondary outcome analysis. The main characteristics of those eligible studies are listed in Table 1 and Table 2. The results of the risk of bias assessment are reported in the Appendix A.

### 3.2. Analysis of the Primary Outcome

#### NIBS versus Sham Treatment

The result of the meta-analysis (*n* = 22 studies, 615 participants) shows that NIBS probably reduced OCD symptoms, with a moderate effect size (SMD = 0.54, 95% CI: 0.26–0.81; Z = 3.84, *p* < 0.01) and heterogeneity (I^2^ = 60.4%, *p* < 0.01) (Figure 2).

Bias test and sensitivity test were conducted to find out the cause of heterogeneity. Two studies [29,31] showed strong heterogeneity. In the view of Gomes et al. [29], the reason for the good effect is that the patients had been sick for fewer years. After the removal of two studies, the results still reached a significant effect size (SMD = 0.37, 95% CI: 0.17–0.58; Z = 3.58, *p* < 0.01), without significant heterogeneity (I^2^ = 26.6%, *p* = 0.13). Egger’s test showed no significant publication bias (*p* = 0.08). The funnel diagram of the bias test is shown in Figure 3.

### 3.3. Subgroup Analysis

#### 3.3.1. Type of Stimulation

In subgroup analysis, the stimulation of different types were considered (Figure 4). The SMD was 0.73 for rTMS (95% CI: 0.38–1.08; Z = 4.04, *p* < 0.01), −0.11 for TBS (95% CI: −0.68–0.46; Z = 0.37, *p* = 0.71), and 0.38 for tDCS (95% CI: −0.04–0.80; Z = 1.75, *p* = 0.08). Compared with sham group, rTMS achieved better therapeutic effects. For TBS and tDCS, there were no significant difference observed.

#### 3.3.2. Stimulation Protocols

In this meta-analysis, LF-rTMS and cTBS were considered as inhibitory stimulation, while HF-rTMS was considered excitatory stimulation. The studies of tDCS were excluded since anodal and cathodal stimulation of tDCS can affect the direction of the induced electric field, not just the “excitatory” or “inhibitory” effects on brain function [47]. In the inhibitory stimulation of the SMA group, SMD was 0.78 (95% CI: −0.09–1.66; Z = 1.75, *p* = 0.08). In the inhibitory stimulation of the OFC group, SMD was 0.24 (95% CI: −0.19–0.67; Z = 1.10, *p* = 0.27). In the excitatory stimulation of the DLPFC group, SMD was 1.13 (95% CI: 0.24–2.01; Z = 2.49, *p* = 0.01). In the inhibitory stimulation of the DLPFC group, SMD was 0.81 (95% CI: 0.26–1.36; Z = 2.87, *p* < 0.01). According to the analytical result, excitatory and inhibitory stimulation of the DLPFC showed some therapeutic effects (Figure 5). If the subgroup contained only one research or the research involved stimulation of two cerebral cortices, they would be excluded from the discussion of results.

### 3.4. Response Rate

According to the results of meta-analysis (*n* = 20 studies, 590 participants), there were more respondents qualified for the response criteria in the NIBS group than in the sham group, with an effect size (RR = 2.26, 95% CI: 1.57–3.25; Z = 4.39, *p* < 0.01). Besides, there was no heterogeneity (I^2^ = 0.0%, *p* = 0.49). In the subgroup analysis based on stimulation type, the RR was 2.48 for rTMS (95% CI: 1.65–3.74; Z = 4.37, *p* < 0.01), 4.67 for tDCS (95% CI: 1.08–20.92; Z = 2.06, *p* = 0.04), and 0.98 for TBS (95% CI: 0.37–2.56; Z = 0.05, *p* = 0.96). Compared with the sham group, the rTMS and tDCS groups achieved higher clinical response rates (Figure 6).

## 4. Discussion

This meta-analysis aimed to investigate the effectiveness of NIBS in reducing OCD symptoms for those patients with refractory OCD. In this paper, a comprehensive comparison was performed between various NIBS methods for refractory OCD. This study also compared the treatment effects of different stimulation types and stimulation protocols to provide better guidance on clinical treatment.

Studies showed that NIBS can be effective in reducing the clinical symptoms of refractory OCD. In the discussion concerning stimulation types, rTMS stimulation was found to be effective in alleviating OCD symptoms and increasing clinical response rates, which is consistent with previous findings [48,49]. For tDCS, there was a significant increase in the clinical response rates. Although the Y-BOCS scores of active groups were reduced, there were no significant differences observed. Silva et al. [46] found that although the Y-BOCS scores remained unchanged at the end of the trial, significant changes were observed at the six-week follow-up. This suggests that tDCS treatment may have a delayed effect. Similarly, prolonged tDCS stimulation also leads to more significant neuroplasticity effects [50]. In addition, several open-label trials [51,52,53,54] and a large case report [55] have also shown certain therapeutic effects of tDCS.

For TBS, there was no significant difference observed in Y-BOCS scores and clinical response rates between the active and sham groups. A reasonable explanation for this finding is that three TBS studies selected the stimulation intensity of 70/80% rMT and 600 pulses per session, but it may be inadequate for refractory OCD patients [41,42,43]. It has been demonstrated that the intensity of the stimulation plays a critical role in determining neuroplasticity [56]. Another plausible explanation is that more sessions over a longer period may be required to obtain a therapeutic effect [57]. Thus, more large randomized controlled trials are required to draw definitive conclusions in the future.

For NIBS therapy, there is an ongoing dispute over stimulation protocols. LF-rTMS decreases cortical excitability by reducing synaptic strength, while HF-rTMS increases cortical excitability [13]. All studies concerning TBS included in this article used cTBS therapy which inhibits cortical excitability [15]. Research showed that 40 s of cTBS depressed motor evoked potential (MEP) for 60 min [15]. In our subgroup study, it was found that both excitatory and inhibitory stimulation of the DLPFC produced some therapeutic effects. According to the relevant research, the hyperactivations in the DLPFC were found in executive function and emotional processing in OCD patients [58]. It seemed to be counterintuitive that the excitatory stimulation of the DLPFC also had therapeutic effects. A plausible explanation for this phenomenon is that the cortical activation is not limited to the stimulated region. Instead, it can be transferred to remote regions through intracerebral networks [59]. It was found that the rTMS at high frequencies has both local and remote effects [60]. 

In contrast, the inhibitory stimulation of the OFC and SMA demonstrated no significant therapeutic effects. SMA is a candidate neurocognitive endophenotype of OCD that may lead to response inhibition dysfunction [61]. Neuroimaging showed SMA hyperactivity in OCD patients [62], and this activity may be associated with insufficient inhibitory control [63]. Likewise, it was found in some studies that OFC hyperactivation was linked to OCD [64,65,66]. Research has demonstrated that there could be significant reductions in gray matter volume in the OFC of OCD patients [67]. The exceptionally high degree of connectivity of the OFC in OCD patients included greater distant connectivity with the subthalamic nucleus and greater local connectivity with the putamen [68]. 

Regarding the approach to analysis, the change in Y-BOCS score was chosen as our primary outcome, not dichotomous data (response rate). This is due to the fact that the dichotomization of continuous data may lead to not only a loss of information and statistical power but also a type I error [69,70]. In addition, due to the different follow-up times of different studies, the medium- and long-term therapeutic effects were not discussed in this paper. In future studies, it will also be worthwhile to provide better guidance on the use of NIBS through neuroimaging and neurophysiological techniques.

## 5. Conclusions

This meta-analysis reveals that NIBS is an effective neurostimulation therapy for refractory OCD. Among them, rTMS produced better results in terms of treatment efficacy and clinical response rates. Besides, tDCS showed an improvement in clinical response rates, while TBS did not appear to show a significant therapeutic effect. In the subgroup analysis, it was discovered that excitatory and inhibitory stimulation of the DLPFC produced better therapeutic effects. Longer follow-up periods and larger sample sizes are required for future randomized controlled studies to investigate better protocols.

## Figures and Tables

**Figure 1 brainsci-12-00943-f001:**
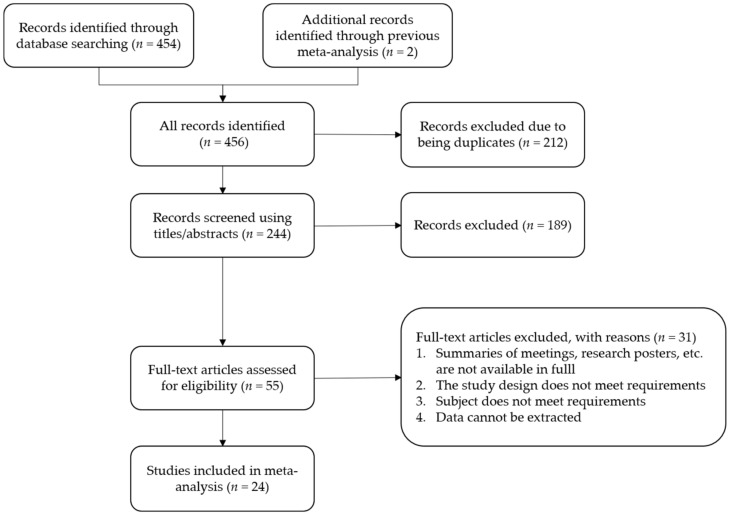
Flowchart of the literature search.

**Figure 2 brainsci-12-00943-f002:**
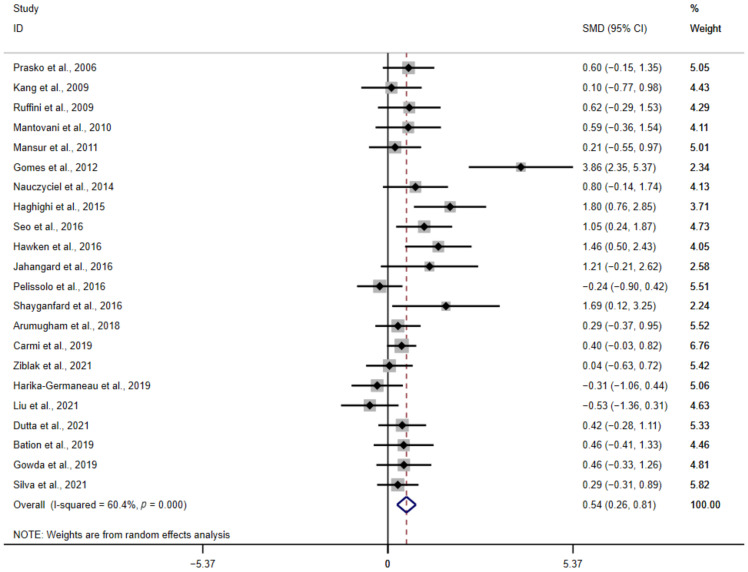
Forest plot of the meta-analysis for the therapeutic effects [23,25,26,27,28,29,30,31,32,33,34,35,36,38,39,40,41,42,43,44,45,46].

**Figure 3 brainsci-12-00943-f003:**
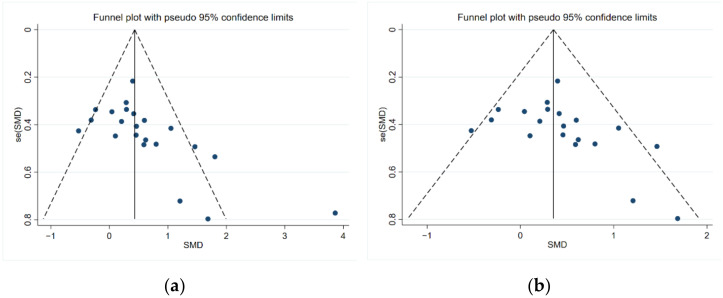
(**a**) Funnel plot of all included studies; (**b**) funnel plot after outlier exclusion.

**Figure 4 brainsci-12-00943-f004:**
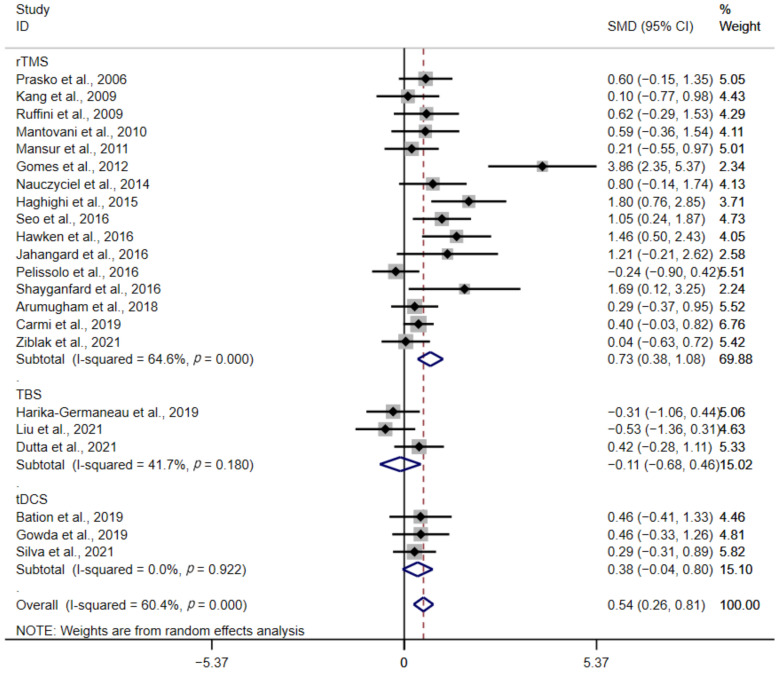
Forest plot of different stimulation types [23,25,26,27,28,29,30,31,32,33,34,35,36,38,39,40,41,42,43,44,45,46].

**Figure 5 brainsci-12-00943-f005:**
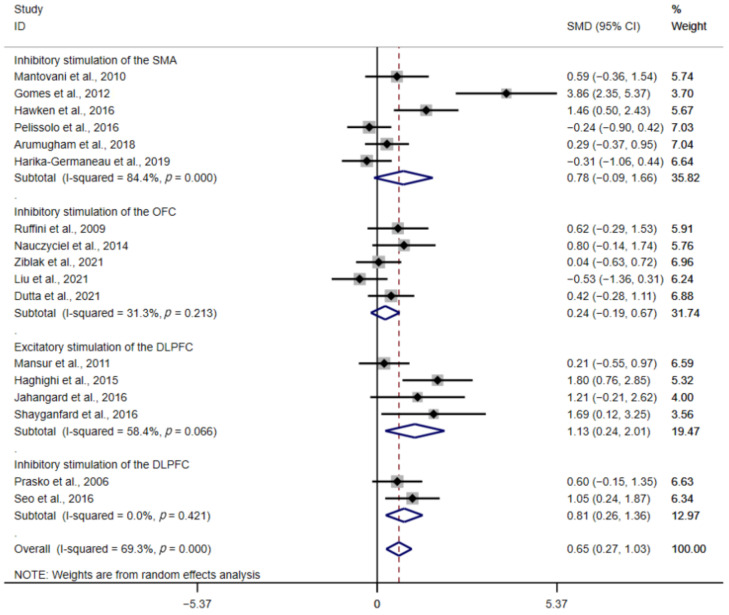
Forest plot of different stimulation protocols [23,26,27,28,29,30,31,32,33,34,35,36,38,40,41,42,43].

**Figure 6 brainsci-12-00943-f006:**
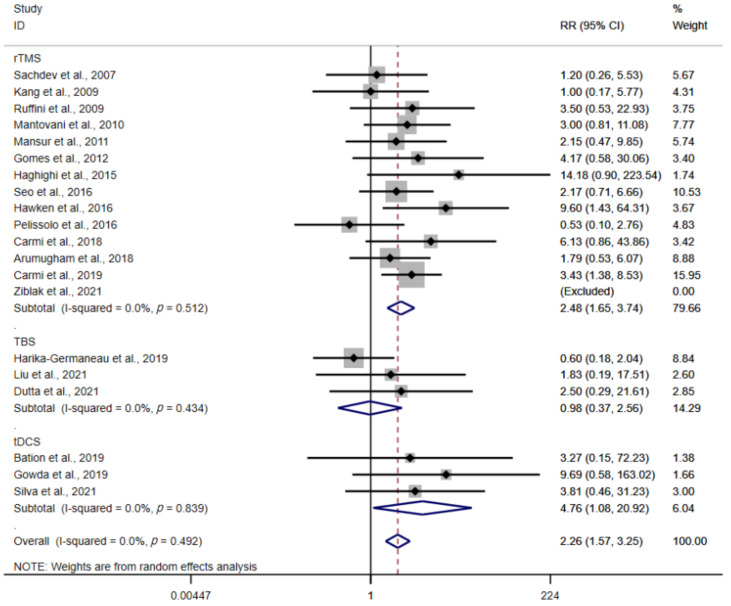
Forest plot of response rates [24,25,26,27,28,29,31,32,33,35,37,38,39,40,41,42,43,44,45,46].

**Table 1 brainsci-12-00943-t001:** Main characteristics of the included studies (rTMS and TBS).

Study	Active		Sham		Sessions	Trial Duration	Parameters				Response’s Definition
	*N*	Age	*N*	Age			Location	Frequency (Hz)	% rMT	Pulses per Session	
**rTMS (*n* = 18)**											
Prasko et al., 2006 [23]	18	28.9 (7.7)	12	33.4 (8.7)	10	2 weeks	L-DLPFC	1	110	1800	NA
Sachdev et al., 2007 [24]	10	29.5 (9.9)	8	35.8 (8.2)	10	2 weeks	L-DLPFC	10	110	1500	40%
Kang et al., 2009 [25]	10	28.6 (12.7)	10	26.2 (10.5)	10	2 weeks	R-DLPFC/SMA	1	110/100	1200	25%
Ruffini et al., 2009 [26]	16	NA	7	NA	15	3 weeks	L-OFC	1	80	NA	25%
Mantovani et al., 2010 [27]	9	39.7 (8.6)	9	39.4 (10.2)	20	4 weeks	SMA	1	100	1200	25%
Mansur et al., 2011 [28]	13	42.1 (11.9)	14	39.3 (13.9)	30	6 weeks	R-DLPFC	10	110	2000	30%
Gomes et al., 2012 [29]	12	35.5 (7.5)	10	37.5 (5.7)	10	2 weeks	SMA	1	100	1200	25%
Nauczyciel et al., 2014 [30]	10	40.0 (NA)	9	39.0 (NA)	10	1 week	R-OFC	1	120	1200	NA
Haghighi et al., 2015 [31]	10	34.9 (5.9)	11	36.6 (4.0)	10	2 weeks	L-DLPFC	20	100	750	35%
Seo et al., 2016 [32]	14	34.6 (9.8)	13	36.3 (12.5)	15	3 weeks	R-DLPFC	1	100	1200	25%
Hawken et al., 2016 [33]	10	33.0 (10.0)	12	34.0 (14.0)	25	6 weeks	SMA	1	110	NA	25%
Jahangard et al., 2016 [34]	5	32.4 (9.0)	5	33.8 (5.8)	10	2 weeks	B-DLPFC	20	100	750	NA
Pelissolo et al., 2016 [35]	20	39.1 (10.4)	16	42.3 (10.6)	20	4 weeks	SMA	1	100	1500	25%
Shayganfard et al., 2016 [36]	5	33.8 (9.6)	5	33.2 (7.9)	10	2 weeks	B-DLPFC	20	100	750	NA
Carmi et al., 2018 [37]	16	36.0 (NA)	14	35.0 (NA)	25	5 weeks	mPFC	20	100	2000	30%
Arumugham et al., 2018 [38]	19	27.7 (7.9)	17	30.7 (10.4)	18	3 weeks	SMA	1	100	1200	35%
Carmi et al., 2019 [39]	47	41.1 (12.0)	47	36.5 (11.4)	29	6 weeks	ACC/mPFC	20	100	2000	30%
Ziblak et al., 2021 [40]	19	41.5 (10.2)	15	36.5 (13.7)	20	2 weeks	R-OFC	1	110	1000	35%
**TBS (*n* = 3)**											
Harika-Germaneau et al., 2019 [41]	14	46.3 (10.1)	14	48.2 (12.9)	30	6 weeks	SMA	50 (cTBS)	70	600	25%
Liu et al., 2021 [42]	12	28.2 (9.8)	11	31.0 (7.5)	10	2 weeks	R-OFC	50 (cTBS)	80	600	25%
Dutta et al., 2021 [43]	18	30.5 (12.4)	15	28.3 (7.4)	10	1 week	L-OFC	50 (cTBS)	80	600	35%

**Table 2 brainsci-12-00943-t002:** Main characteristics of the included studies (tDCS).

tDCS Study (*n* = 3)	Active		Sham		Sessions	Trial Duration	Parameters				Response’s Definition
	*N*	Age	*N*	Age			Anode Location	Cathode Location	Current	Session Duration	
Bation et al., 2019 [44]	10	44.8 (19.9)	11	41.2 (11.9)	10	1 week	cerebellum	OFC	2 mA	20 min	35%
Gowda et al., 2019 [45]	12	30.83 (5.9)	13	25.9 (5.2)	10	1 week	SMA	supra-orbital area	2 mA	20 min	35%
Silva et al., 2021 [46]	22	38.4 (11.0)	21	36.9 (12.2)	20	4 weeks	deltoid	SMA	2 mA	30 min	35%

L, left; R, right; B, bilateral; DLPFC, dorsolateral prefrontal cortex; SMA, supplementary motor area; OFC, orbitofrontal cortex; mPFC, medial prefrontal cortex; ACC, anterior cingulate cortex; rMT, resting motor threshold; Response’s definition, Y-BOCS score’s decrease percentage; NA, not applicable.

## Data Availability

Data are available upon reasonable request from the corresponding author.

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
