# Peer review of "Efficacy of Non-Invasive Brain Stimulation for Refractory Obsessive-Compulsive Disorder: A Meta-Analysis of Randomized Controlled Trials"

_brainsci, 2022, doi:10.3390/brainsci12070943_

Round 1

Reviewer 1 Report

A very meaningful topic and well analyzed meta data in NIBS area. The statistic methods are presented in details and the analysis is in depth. Very well organized frame for a Meta analysis, the discussion and conclusion are both of interesting and adequate. some mild English errors should be rechecked and corrected. For example: in 2.3 Outcome Measures, at line 82, " We will b calculated by using baseline ...", I believe  "  We" is a typo,  could be" It" or "They", refer to the data missed. 

Author Response

Thank you very much for your comments and suggestions on this article.

We regret there were problems with the English. The paper has been carefully revised by a professional language editing service to improve the grammar and readability.

Reviewer 2 Report

Publication entitled " Efficacy of Non-invasive Brain Stimulation for Refractory Obsessive-compulsive Disorder: A Meta-analysis of Randomized   Controlled Trials" represents a

systematic review of clinical studies of refractory OCD in which noninvasive brain neuromodulation was applied.

The review includes rTMS, TBS and tDCS, which are indeed the most widely used and best known neuromodulation techniques to date.

An additional threshold of "acceptance" of the study for this analysis were the requirements that only RCTs and minimal 10 sessions –treatments were accepted, which provides methodological adherence, but at the same time relevance fo real life situations.

The review included a total of 24 studies, of which 18 were rTMS, and 3 each for TBS and tDCS.

Although the analysis of stimulation outcomes, measured by the effects of stimulation on CSO symptoms through the Yale - Brown Obsessive Compulsive Scale, is presented cumulatively for all studies, I personally consider it important to consider the effects of each method individually (shown in Figure 4).

I would also like to mention that some references in the field of neuromodulation certainly need to be replaced, since they are obsolete.

So I suggest that reference no. 14:

Lefaucheur, J.P .; Andre-Obadia, N .; Antal, A .; Ayache, S.S .; Baeken, C .; Benninger, D.H .; Cantello, R.M .; Cincotta, M .; de Carvalho, M .; De Ridder, D .; et al. Evidence-based guidelines on the therapeutic use of repetitive transcranial magnetic stimulation (rTMS). Clin Neurophysiol 2014, 125, 21502206, doi: 10.1016 / j.clinph.2014.05.021) replace with reference from the same author, but more recent

Lefaucheur, J. P. et al. Evidence-based guidelines on the therapeutic use of repetitive transcranial magnetic stimulation (rTMS): An update (2014-2018). Clin. Neurophysiol. 131, 474–528 (2020).

Author Response

Thanks a lot for your comments and suggestions.

Regarding the issue of outdated references you mentioned, we have done a careful review of the references and replaced them accordingly.

Also, we agree with you about the importance of considering the effects of each method individually. In the analysis of the secondary outcome of clinical response rate, we also focused the analysis on the effect of each approach (shown in Figure 6).

Reviewer 3 Report

This is a summary article about different types of non-invasive brain stimulation in the treatment of the obsessive-compulsive disorder. This investigation is somewhat interesting; however, there are still many questions that need to be addressed before the manuscript can be published.

Questions:

1.       The discussion section has many problems, such as not being clearly organized and not reflecting the authors' main findings in this investigation.

2.       The authors mention, "The left DLPFC of OCD patients had a higher fractional amplitude of low-frequency fluctuation", how does this relate to your study?

3.       Many references are missing in the manuscript.

4.       The references [58] do not match those described in the manuscript.

5.       The description in the reference [65] is contrary to the original study.

Author Response

Thanks a lot for your comments and suggestions.

  1. In response to the unclear organization of the discussion section, we did further discussions and made appropriate modifications to make the structure of the article clearer and easier to read.
  2. As for the sentence "the left DLPFC of OCD patients had a higher fractional amplitude of low-frequency fluctuation". What we want to express here is that there is hyperactivations in the DLPFC in OCD patients. It may not be clearly expressed in the expression, and we have replaced the relevant literature and expressions (shown in the reference [60]).
  3. Regarding the issue of missing references in the manuscript you describe, we feel very sorry. We have done a detailed review and supplemented as much as possible.
  4. Regarding reference [58], our expressions may not be very accurate. We have replaced the reference with a more appropriate expression (shown in the reference [60]).
  5. Regarding reference [65], due to our negligence, we realized that there are some misunderstandings about the meaning of the original text after your reminder. At present, we have deleted the relevant literature in the revised version.
  6. Regarding the problem of English expression, we regret there were problems with the English. The paper has been carefully revised by a professional language editing service to improve the grammar and readability.
  7. We have also carefully checked the problem that the method may be insufficiently described, and the registration number of this meta-analysis on the PROSPERO is displayed (shown in line 61).

Reviewer 4 Report

the article is well built and discretely detailed , methodologically correct.

I only suggest to the author the opportunity to outline that the results and conclusions related to TBS and tDCS  treatments have poor relevance due to the scarce numerosity of works (3 and 3 works respectively) and subjects  (40 and 44sbjcts) . Moreover, for TBS it is not specified if the treatment was  cTBS or iTBS .

For these reason the conclusion that rTMS is more efficacious than TBS or tDCS should be expressed more cautiously.

Author Response

Thanks a lot for your comments and suggestions.

  1. Regarding the issue that there are few RCT literatures related to TBS and tDCS treatment, in the revised version, the results of relevant open-label trials and case report were presented in the discussion section to give readers a more comprehensive consideration (shown in line 206-207).
  2. Regarding the treatment of TBS types, the use of cTBS or iTBS have already mentioned in the Table 1. Continued. In addition, the type of TBS treatment is mentioned again in the discussion section in this revision (shown in line218-219).
  3. In the conclusion that rTMS is more efficient than TBS or tDCS, we have replaced the more cautious expressions (shown in line 160). In the alternative expression, we emphasize that the advantage of the treatment effect is compared to the sham treatment, not to the other two treatments.

Round 2

Reviewer 3 Report

The authors have addressed all my questions in a satisfactory way.